# Primary Reason for Drinking Among Current, Former, and Never Flushing College Students

**DOI:** 10.3390/ijerph16020211

**Published:** 2019-01-13

**Authors:** Sarah Soyeon Oh, Yeong Jun Ju, San Lee, Eun-Cheol Park

**Affiliations:** 1Institute of Health Services Research, Yonsei University, Seoul 03722, Korea; sarahoh@yuhs.ac (S.S.O.); joomeon@gmail.com (Y.J.J.); thismountain@yuhs.ac (S.L.); 2Department of Public Health, Graduate School, Yonsei University, Seoul 03722, Korea; 3Department of Psychiatry, Yonsei University College of Medicine, Seoul 03722, Korea; 4Department of Preventive Medicine, Yonsei University College of Medicine, Seoul 03722, Korea

**Keywords:** alcohol flush reaction, flushing, aldehyde dehydrogenase 2, ALDH2, college drinking

## Abstract

Alcohol consumption among individuals who experience a flushing response (reddening of the face, nausea, dizziness, headache, anxiety, and increased heartbeat) can result in serious health problems. However, studies on reasons for drinking among flushers, especially in the college context, are limited. Thus, this study investigated the association between primary reason for drinking and alcohol use among a nationally representative sample of current, former, and never flushing college students. The aim was to measure whether college students with current or former experience of facial flushing have different primary reasons for drinking compared to students with no experience of facial flushing. We surveyed and analyzed the data of 4590 students in a nationally representative sample of 82 colleges in South Korea. Multiple regression analysis was used to identify the association between primary reason for drinking and alcohol intake. Alcohol intake was measured using the Alcohol Use Disorders Identification Test (AUDIT). Among 1537 current (33.5%), 152 former (3.3%), and 2901 (63.2%) never flushers, mean AUDIT scores were 7.715 ± 5.434, 11.039 ± 6.405, and 10.465 ± 5.779, respectively. Current flushers had significantly higher AUDIT scores when drinking for pleasure (β = 2.696, *p* < 0.0001) or stress/depression (β = 2.578, *p* < 0.0001). Primary reasons for drinking were not associated with alcohol intake for former flushers. Never flushers had significantly higher AUDIT scores when drinking for pleasure (β = 2.696, *p* < 0.0001), stress/depression (β = 2.578, *p* < 0.0001), or boredom (β = 0.740, *p* = 0.029) than peer pressure. Our results suggest that former and never flushers consume higher amounts of alcohol on average than never flushers. For current flushers, drinking for pleasure or stress/depression may increase alcohol intake, while for never flushers, drinking for pleasure, stress/depression, as well as boredom may have the same effect.

## 1. Introduction

Alcohol consumption among college students contributes to an average of approximately 1825 deaths (e.g., from homicides, motor-vehicle crashes, and suicides), 97,000 sexual assaults, and 600,000 injuries in the United States each year [1]. In South Korea, college drinking is a major social problem; 60.4% of college students report to binge drinking at least once a month, and 10.8% of deaths among college students are related to drinking [2]. Furthermore, around 80% of college students experience physical discomfort from drinking, 43.7% have blackouts, and more than 50% of colleges report to suffering from problems that emerge from student drinking like vandalism and campus violence [2].

Consequently, a large body of research has identified reasons for drinking and alcohol intake among college students. Such reasons include motives to maintain or increase positive affective states [3], social outcome expectancies [4], and amelioration of negative emotions like stress, depression, and/or tension [5]. Primary reasons for drinking can also emerge from the college environment in the form of peer pressure [6], campus alcohol policy [7], and direct offers to use alcohol [8].

However, few studies have compared reason for drinking and alcohol intake according to flushing status. This is alarming as flushing upon alcohol intake is a common health concern among East Asians. Brooks and colleagues found that approximately 36% of Korean, Chinese, and Japanese individuals experience facial flushing [9] while Takeshita and colleagues found that nearly half of their Japanese sample manifested symptoms of facial flushing and palpitation upon consuming alcohol [10].

Flushing occurs among individuals who inherit a deficiency in the acetaldehyde-breaking enzyme, aldehyde dehydrogenase 2 (ALDH2) [9]. Upon consuming even small amounts of alcohol, flushers can experience a reddening of the face, nausea, dizziness, headache, anxiety, and increased heartbeat [11]. Such symptoms signify a buildup of toxic acetaldehyde in the body; toxic acetaldehyde that has been classified as a group one human carcinogen by the International Agency for Research on Cancer of the World Health Organization [12]. Malignancies associated with flushing include esophageal [13], gastric [14,15,16,17], colorectal [18,19,20], and stomach [21] cancers. Furthermore, researchers have stated that if moderate or heavy drinking flushers decreased their alcohol intake, more than 50% of esophageal squamous cell carcinomas could be prevented in East Asian samples [9].

Fortunately, because of the unpleasant and noticeable consequences of drinking alcohol for flushers, flushers consume less alcohol than non-flushers. In many studies, it has been noted that flushers consume lower amounts of alcohol than non-flushers with respect to drinking frequency, binge drinking, and number of drinks per drinking session [22]. A 2006 study of Chinese and Korean college students also revealed that flushing can be a protective mechanism as flushers are less likely to drink heavily in one drinking session or have alcohol-induced blackouts [23]. This may be because college students often intervene and encourage flushers to stop drinking alcohol, especially if they are a female or close friend [24]. However, in another study of 1080 Chinese undergraduate students conducted in 2013, around 20% of male respondents stated that they would “encourage a flushing man to drink more” and “do not see flushing as an indicator to stop or slow down drinking” [24].

Subsequently, primary reasons for drinking are likely to differ among flushers and non-flushers. In a 1997 study of Japanese college students, it was found that while non-flushers are likely to enjoy consuming alcohol at home, with intimate friends, or at club parties, whereas flushers are more likely to consume alcohol when pressured to drink, especially by non-flushers, at social events [25].

Given the limited literature on reasons for drinking and alcohol intake among current, former, and never flushing college students, our study asks two questions. Firstly, do the same primary reasons for drinking and alcohol use apply to flushers and non-flushers? Secondly, how do the primary reasons for drinking and alcohol use differ among current flushers and former flushers? We hypothesize that different primary reasons for drinking and alcohol use will apply to flushers and non-flushers due to the unpleasant and noticeable consequences of drinking for flushers. According to previous studies, flushers are more likely to drink because of social pressures than enjoyment, whereas non-flushers have a range of reasons for drinking including positive, social, and emotional [5]. We also hypothesize that the primary reasons for drinking and alcohol use among current flushers and former flushers will differ because former flushers are students who continued to drink despite experiencing the flushing response; and therefore, are likely to associate drinking alcohol with pleasure, and/or alleviation of stress/depression.

The present study examines the association between primary reasons for drinking and alcohol intake among current, former, and never flushing students in a nationally representative sample of college students in South Korea. It also investigates the factors that strengthen and weaken this association.

## 2. Materials and Methods

### 2.1. Study Sample and Data

In the 2017 national statistics published by the Korean Educational Development Institute on college students, we found that 1,951,940 students (4-year courses of study: 1,506,745; 2-year courses of study: 445,195) are enrolled in 356 colleges (4-year courses of study: 195, 2-year courses of study: 161) in South Korea. From these colleges, we excluded 23 that had fewer than 500 administered students or were located in the remote island of Jeju. Of the remaining 333 colleges, we randomly selected 85 colleges for our survey analyses. During the recruitment process, three four-year colleges declined to participate in our survey for religious reasons, decreasing the total number of colleges in our investigation to 82 (4-year: 54; 2-year: 28). From these colleges, we stratified a proportionately representative sample of undergraduate students to match national statistics (Seoul/Incheon/ Gyeonggi, Gangwon, Daejeon/Chungjeong, Gwangju/Jeolla, Daegu/Gyeongbuk, Busan/Ulsan/ Gyeongnam) (see Appendix A).

Data were collected via face-to-face surveys with interviewers randomly selecting students passing by each campus’s department buildings. Data collectors were instructed to survey around 60 students from each campus; three males and three females from 10 different majors. Teams of collectors were trained for consistency by Gallup and researchers of our investigation; each question of the questionnaire was required to be administered orally in a face-to-face manner, at an enclosed space like a café or lecture room.

In total, 5000 students completed our survey instrument. The response rate was 68.7%, with the total number of approached participants being 7278. A financial incentive of 10,000 Korean Won (equivalent to around 9 US dollars) was given to each participant upon completion of the 14-page survey instrument. For the purpose of this investigation, we excluded 211 students who reported never consuming a sip of alcohol in their entire lives, 199 students who reported not consuming a single drink in the last 12 months, and 43 students who reported only consuming alcohol during special traditional Korean events.

The survey instrument asked students a number of questions about their drinking behavior, health, and thoughts on campus-alcohol policy. Whenever possible, the instrument included alcohol-related questions that had been previously given in other international, national or large-scale epidemiological studies including the Harvard College Alcohol Study, the Korea National Health and Nutrition Examination Survey (KNHANES), and the Korea Youth Risk Behavior Web-Based Survey (KYRBS).

Following the standards of the Korea Centers for Disease Control & Prevention, a standard drink was defined as the amount of alcohol contained in one standard drinking glass of alcohol (approximately 8 grams of pure alcohol), equivalent to: 1 shot of soju, 1 glass of bottled beer, 2/3 of a canned beer, 1/2 glass of draft beer, 1/2 bowl of makgeolli (rice wine), 1/2 glass of wine, 1 glass of whiskey, 1 shot of cheongju (refined rice wine), 1 shot of herbal liquor, 1 shot of fruit wine, or a 3/5 glass of mixed liquor (soju+beer).

All subjects gave their informed consent for inclusion before they participated in the study. The study was conducted in accordance with the Declaration of Helsinki, and the protocol was approved by Yonsei University Health System’s Institutional Review Board (Y-2017-0084).

### 2.2. Measures

#### Outcome Variable

In this study, alcohol use was measured through the Alcohol Use Disorders Identification Test (AUDIT) scale and selected as the outcome variable. The AUDIT questionnaire was given in its original format, consisting of ten questions related to frequency of drinking, number of drinks per session, frequency of heavy drinking, impaired control following drinking, morning-after drinking, and feelings of guilt upon drinking, frequency of blackouts, alcohol-related injuries, and family concern over drinking. While total scores can range from 0 to 40, the World Health Organization states that a score of 0–7 indicates low-risk of alcohol-related problems and harm, 8–15 moderate risk, 16–19 high-risk, and 20 or more almost certainly representative of alcohol dependence [26]. As a screening criteria of alcoholism in Korea, the AUDIT has a sensitivity and specificity of 96.9% and 87.1% for determining “problem drinking”, a sensitivity and specificity of 89.5% and 79.5% for determining “moderate risk”, and a sensitivity and specificity of 85.7% and 93.3% for determining “alcohol dependence”, respectively, depending on cut-off scores [27].

### 2.3. Primary Reason for Drinking

Primary reason for drinking was measured via individual answers to the question, “In the last 12 months, what was your primary reason for drinking alcohol?” Respondents were required to select one primary reason for their drinking in the last 12 months, ranging from the following options: “peer pressure,” “pleasure,” “stress/depression,” “boredom,” and “other”. “Other” responses included the following: “force of habit,” “birthday,” “anniversary,” “to sleep.”

### 2.4. Current, Former, and Never Flushing

Flushing was measured through the questionnaire created by Yokoyama and Omori, which asks the following questions to determine past and present ALDH2 heterozygotes: (1) Do you have a tendency to develop facial flushing immediately after drinking a glass (about 180 mL) of beer? (2) Did you have the tendency in the first one or two years after you started drinking? Individuals who replied “yes” to question 1 were determined current flushers; individuals who replied “yes” to the second question were determined former flushers, and individuals who replied “no” to both questions were determined never flushers. This questionnaire was rated to have a high sensitivity rate of 90.1% and specificity rate of 88.0% [28] in classifying respondents into never, former, and current sufferers of ALDH2 deficiency. Although created for a Japanese sample, the questionnaire was validated for Korean samples and found to have a relatively lower sensitivity rate of 78.9% and specificity rate of 82.1% [29].

### 2.5. Statistical Analysis

A series of three regression analyses were conducted for our investigation. First, frequencies and mean AUDIT scores were calculated for subjects according to sex, year level, major, grade point average (GPA), allowance, living status, smoking, underage drinking experience, and number of sororities/clubs through analyses of variance. To examine the association between primary reason for drinking and alcohol use, multiple regression analysis were performed, after controlling for the following confounders: sex, year level, major, GPA, allowance, living status, smoking, underage drinking experience, and number of sororities/clubs. Lastly, an examination of primary reasons for drinking by flushing status, by gender was conducted for our subgroup analyses. Multiple regression analyses were used to test this interaction, while controlling for year level, major, grade point average (GPA), allowance, living status, smoking, underage drinking experience, and number of sororities/clubs. The calculated p-values in this study were considered significant if lower than 0.05. All analyses were performed using SAS software, version 9.4 (SAS Institute, Cary, NC, USA).

## 3. Results

Table 1 shows the general characteristics of the study sample. Among 1537 current (33.5%), 152 former (3.3%), and 2901 (63.2%) never flushers, mean AUDIT scores were 7.715 ± 5.434, 11.039 ± 6.405, and 10.465 ± 5.779, respectively. Drinking due to peer pressure (33.1%) was the most popular primary reason for drinking among current flushers, while drinking for pleasure (40.1%) was the most popular primary reason for drinking among never flushers.

Table 2 shows the results of the linear regression analysis performed to investigate the association between various factors and AUDIT score among college students, categorized by current flushers, former flushers, and never flushers. Current flushers had significantly higher AUDIT scores when drinking for pleasure (β = 2.696, *p* < 0.0001) or stress/depression (β = 2.578, *p* < 0.0001). Having a grade point average below 3.0 (β = 1.125, *p* = 0.017), having higher amounts of allowance or spending money (Q2 β = 0.934, *p* = 0.004; Q3 β = 1.641, *p* < 0.0001; Q3 β = 1.985, *p* < 0.0001), living alone/flatting (β = 1.362, *p* < 0.0001), having underage drinking experience (β = 0.671, *p* = 0.010), and being in two or more sororities (β = 1.571, *p* = 0.000) were also associated with higher AUDIT scores.

No primary reasons for drinking were associated with higher AUDIT scores for former flushers. However, having a grade point average below 3.0 (β = 4.356, *p* = 0.028), being in the highest allowance quartile (β = 4.919, *p* = 0.000), and having underage drinking experience (β = 4.163, *p* < 0.0001) were associated with increased AUDIT scores.

Never flushers had significantly higher AUDIT scores when drinking for pleasure (β = 2.696, *p* < 0.0001), stress/depression (β = 2.578, *p* < 0.0001), or boredom (β = 0.740, *p* = 0.029) than peer pressure. Having a grade point average below 3.0 (β = 1.099, *p* = 0.002), having higher amounts of allowance (Q2 β = 0.694, *p* = 0.005; Q3 β = 1.872, *p* < 0.0001; Q3 β = 2.439, *p* < 0.0001), living status (living alone/flatting β = 1.279, *p* < 0.0001; college dorm β = 0.868, *p* = 0001), and having underage drinking experience (β = 2.573, *p* < 0.0.001) were associated with higher AUDIT score. Such statistically significant trends were not found in the logistic regression analysis investigating the association between various factors and a one-unit increase in the AUDIT (see Appendix A).

Figure 1 shows the results of the subgroup analysis for the association between primary reasons for drinking and alcohol intake, according to sex. Among current flushers, males had higher AUDIT scores when drinking for stress/depression (β = 2.635, *p* < 0.0001) and females had higher AUDIT scores when drinking for pleasure (β = 2.741, *p* < 0.0001). Among former flushers, both males (β = 3.186, *p* = 0.096) and females (β = 2.086, *p* = 0.3346) had higher AUDIT scores when drinking due to boredom, however, this association was not statistically significant. Among never flushers, both males (β = 2.483, *p* < 0.0001) and females (β = 3.842, *p* < 0.0001) had higher AUDIT scores when drinking due to pleasure.

## 4. Discussion

In our investigation, former flushers had the highest average alcohol consumption score among current, former, and never flushing individuals. With continued drinking, current flushers can have diminishing symptoms over time and become former flushers; alcohol flushing diminishes in intensity due to the development of tolerance to acetaldehydemia by higher-risk persons with a long or heavy drinking history [28]. However, studies show that former flushers are as equally at risk of various aerodigestive tract cancers as current flushers, if not in greater risk due to continued drinking [30]. In a case-control study comparing 96 Japanese men with oral and pharyngeal squamous cell carcinomas, it was found that current or former flushers have greater risk of both cancers than current flushers alone, especially if alcoholic [28].

One possible explanation for the emergence of former flushers is that individuals who flush often attempt to overcome the social embarrassment of flushing by continuing to drink [31]. However, our study showed no association between primary reasons for drinking (e.g., peer pressure) and increased alcohol intake among former flushers.

According to a study of Asian-Americans and Pacific Islanders, pleasure is not a reaction that is usually associated with reason for drinking among flushers. Because of the unpleasant flushing reaction, individuals with the flushing reaction are likely to consume less or no alcohol, and have reduced risk for alcoholism [32]. On the contrary, ALDH2-deficient university students are likely to drink heavily when faced with peer pressure [9].

However, in our study, it was found that female flushers drink more for pleasure than peer pressure. The difference may be cultural; from 2005 to 2015, there has been a 10% increase in the monthly drinking rate of South Korean women between the ages of 19 and 29 [33]. The difference may also suggest that for Korean college students, flushing does not result in lower consumption of alcohol.

The associations between primary reasons for drinking and alcohol intake among current and never flushing college students were the main findings of our study. While both groups had greater alcohol intake if primary reasons for drinking included pleasure or stress/depression, drinking due to boredom was only associated with increased alcohol intake among never flushers. As drug-makers have associated flushing with displeasure and incorporated acetaldehyde dehydrogenase inhibition to develop effective treatments for alcoholics, more research must be done regarding this finding [34,35].

Furthermore, among current flushers, males had higher AUDIT scores when drinking due to stress/depression while females had higher AUDIT scores when drinking for pleasure. Among never flushers, both males and females had higher AUDIT scores when drinking for pleasure. In previous studies, popular reasons for drinking among college students include pleasure and social reasons; however, stress was only a primary reason for drinking among 2.1% of college students [36].

This study has several limitations that should be considered when interpreting results. First, our study is cross-sectional in design; thus, caution should be exercised in interpreting causality between primary reason for alcohol consumption and alcohol intake. Furthermore, the Yokoyama and Omori questionnaire for determining ALDH2 deficiency are limited in reliability and validity relative to clinical evaluations of these symptoms by a healthcare professional. A more reliable method of evaluating the ALDH2 genotype would be combined use of the self-reported facial flushing questionnaire and an ethanol patch test, as used by Ishibashi and colleagues [37]. Also, given that the sensitivity and specificity of the Korean version of the questionnaire was not as high as the Japanese version, the appropriateness of incorporating this measure in our investigation of Korean individuals should also be questioned.

Second, there are not enough previous studies with regard to a nationally representative sample of Koreans when it comes to measuring flushing and its effect on drinking behavior/related problems. It is difficult to see whether the values we calculated are similar to that of the statistics found in previous studies for Koreans, especially for the college students’ age group.

Third, various biases may have emerged from our sampling and surveying methods; because college students in South Korea drink large amounts of alcohol relative to adults, different patterns are likely to emerge in an adult sample. Likewise, a small number of Christian colleges that were originally in our sample declined our request for participation because of their principles regarding abstaining from drinking and thus, had to be replaced with non-Christian colleges. Because of the face-to-face method that we employed for accuracy of obtaining responses to complicated questions, there may have been response biases, relative to social desirability. The majority of questions in our survey instrument required students to think about their drinking behaviors in the last 12 months or so, which likely resulted in recall bias. Finally, although we included numerous lifestyle covariates as potential confounders, the limited nature and number of questions in our instrument made it difficult in other confounding variables, relative to health, socio-demographics, gene-environment, and lifestyle, to be measured and controlled.

Lastly, our survey measure only questioned the “primary reason” for drinking among students in the last 12 months. However, over a 12 month period, drinkers are likely to drink for a variety of primary reasons and this should be considered when interpreting our results.

Despite these limitations, our study also has several strengths. Few studies have measured ALDH2 deficiency for a nationally representative sample in South Korea or taken an epidemiological approach to see the association between primary reasons for drinking and alcohol intake among current, former, and never flushing college students.

## 5. Conclusions

Our study has found certain risk-groups in the college sample including former flushers, who primary drink due to peer pressure and/or social embarrassment, male flushers who consume alcohol due to stress/depression, female flushers who consume alcohol due to pleasure, and never flushers who primarily consume alcohol due to pleasure, stress/depression, and boredom. However, more research is required to understand why female flushers consume alcohol due to pleasure despite the disagreeable side effects of drinking in a college context. More research must also be conducted on the primary reasons for drinking among former flushers as they are in the greatest danger of developing detrimental health effects among current, former, and never flushers. Researchers, educators, and policy-makers are encouraged to further investigate, and target such students when creating campus alcohol initiatives and education programs to alleviate these problems.

## Figures and Tables

**Figure 1 ijerph-16-00211-f001:**
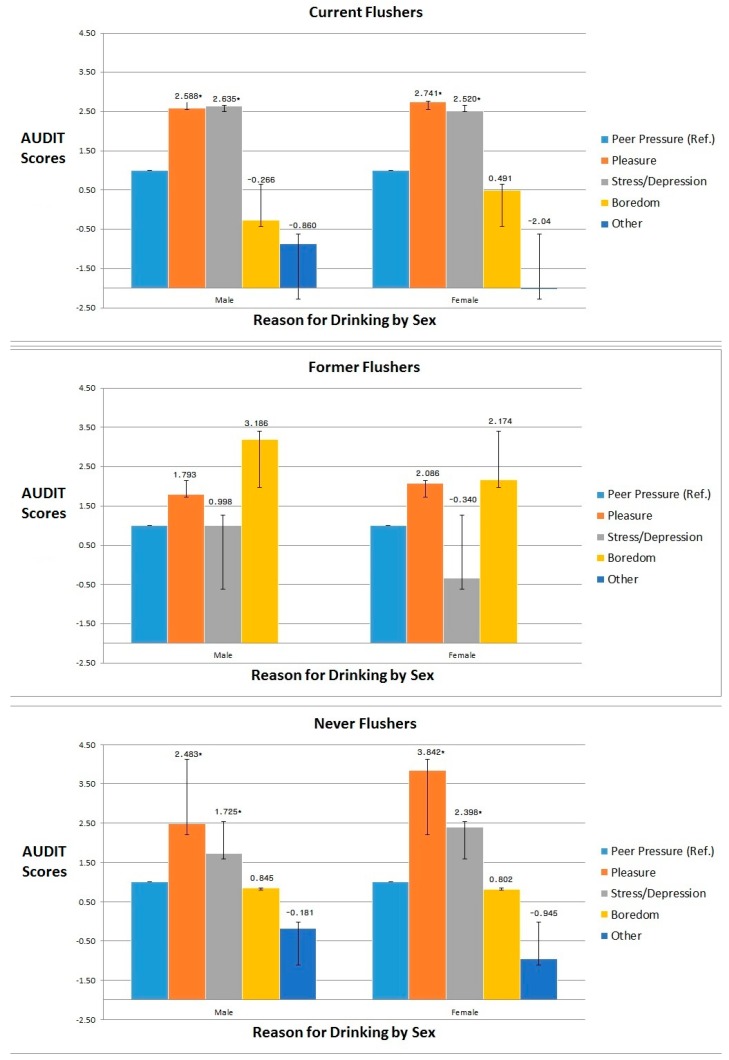
Subgroup analysis for the association between primary reason for drinking and Alcohol Use Disorders Identification Test scores (AUDIT), according to sex. * Statistically significant results (*p* < 0.05).

**Table 1 ijerph-16-00211-t001:** General characteristics of the study population by current, former, and never flushing status.

Variables	Alcohol Use Disorders Identification Test (AUDIT) Scores											
	Current Flusher				Former Flusher				Never Flusher			
	n	%	Mean	SD	n	%	Mean	SD	n	%	Mean	SD
**Primary Reason for Drinking**												
Peer Pressure	508	33.05	6.104	5.218	34	22.37	8.971	6.534	704	24.27	8.227	5.022
Pleasure	464	30.19	9.358	5.247	40	26.32	12.9	6.054	1163	40.09	12.095	5.811
Stress/Depression	346	22.51	8.821	5.657	56	36.84	10.554	6.608	648	22.34	10.685	5.848
Boredom	198	12.88	6.404	4.483	22	14.47	12.091	5.485	345	11.89	9.391	5.141
Other	21	1.37	4.524	3.558	0	0	-	-	41	1.41	8.171	5.928
**Sex**												
Male	725	47.17	8.254	5.341	74	48.68	11.649	6.123	1460	50.33	10.556	5.401
Female	812	52.83	7.234	5.474	78	51.32	10.462	6.648	1441	49.67	10.372	6.139
**Year Level**												
1	483	31.42	7.594	5.47	50	32.89	12.96	6.55	927	31.95	10.385	5.791
2	470	30.58	8.223	5.526	56	36.84	10.661	7.242	954	32.89	10.65	5.859
3	277	18.02	8.072	5.36	23	15.13	9.13	3.468	480	16.55	10.367	5.666
≥4	307	19.97	6.805	5.196	23	15.13	9.696	5.329	540	18.61	10.361	5.724
**Major**												
Humanities and Social Sciences	734	47.76	7.691	5.496	79	51.97	11.342	6.145	1356	46.74	10.321	5.78
Engineering/Natural Sciences	624	40.6	7.825	5.372	58	38.16	10.828	6.837	1199	41.33	10.52	5.685
Liberal Arts	179	11.65	7.43	5.412	15	9.87	10.267	6.352	346	11.93	10.838	6.089
**Grade Point Average**												
≥4.0	239	15.55	7.31	4.98	13	8.55	7.846	4.828	414	14.27	10.408	5.626
3.5–4.0	559	36.37	7.161	5.264	54	35.53	10.463	5.878	1016	35.02	10.066	5.824
3.0–3.5	536	34.87	8.019	5.601	58	38.16	11.034	7.051	1035	35.68	10.309	5.599
≤3.0	203	13.21	8.916	5.74	27	17.76	13.741	5.887	436	15.03	11.817	6.057
**Allowance**												
Q1 (Low)	596	38.78	6.525	4.788	54	35.53	9.481	6.002	994	34.26	9.046	5.383
Q2	396	25.76	7.697	5.559	45	29.61	10.267	6.214	758	26.13	10	5.315
Q3	281	18.28	8.651	5.628	25	16.45	12.04	6.617	600	20.68	11.49	6.06
Q4 (High)	264	17.18	9.432	5.776	28	18.42	14.393	6.196	549	18.92	12.554	5.966
**Living Status**												
Family Home	814	52.96	7.022	5.222	82	53.95	10.829	6.348	1570	54.12	9.71	5.641
Living Alone/Flatting	369	24.01	9.247	5.976	43	28.29	11.791	7.21	747	25.75	11.857	6.007
College Dorm	354	23.03	7.712	4.989	27	17.76	10.481	5.228	584	20.13	10.714	5.498
**Smoking Status**												
Current Smoker	323	21.01	9.498	5.365	44	28.95	12.568	6.739	703	24.23	12.376	5.667
Past Smoker	86	5.6	10.721	6.001	10	6.58	12.3	5.187	118	4.07	12.534	5.724
Non-Smoker	1128	73.39	6.975	5.211	98	64.47	10.224	6.271	2080	71.7	9.701	5.642
**Underage Drinking**												
Yes	752	48.93	8.638	5.598	91	59.87	12.978	6.36	1431	49.33	12.151	5.864
No	785	51.07	6.831	5.122	61	40.13	8.148	5.326	1470	50.67	8.823	5.196
**Number of sororities/clubs**												
None	767	49.9	7.254	5.237	79	51.97	10.203	6.884	1512	52.12	10.458	5.747
One	611	39.75	7.866	5.343	54	35.53	11.907	5.506	1121	38.64	10.263	5.727
Two or more	159	10.34	9.358	6.328	19	12.5	12.053	6.561	268	9.24	11.343	6.109
**Total**	1537	33.49	7.715	5.434	152	3.31	11.039	6.405	2901	63.2	10.465	5.779

**Table 2 ijerph-16-00211-t002:** Results of multiple regression analysis to investigate association between factors and Alcohol Use Disorders Identification Test (AUDIT) scores.

Variables	AUDIT Scores								
	Current Flusher			Former Flusher			Never Flusher		
	β	SE	*p*-Value	β	SE	*p*-Value	β	SE	*p*-Value
**Primary Reason for Drinking**									
Peer Pressure	Ref.			Ref.			Ref.		
Pleasure	2.696	0.319	<0.0001	0.777	1.382	0.574	3.125	0.248	<0.0001
Stress/Depression	2.578	0.344	<0.0001	0.244	1.219	0.841	1.984	0.282	<0.0001
Boredom	0.161	0.411	0.695	1.965	1.514	0.194	0.740	0.339	0.029
Other	−1.510	1.091	0.167	−	−	−	−0.302	0.826	0.715
**Sex**									
Male	Ref.			Ref.			Ref.		
Female	−0.058	0.277	0.835	0.650	1.076	0.546	0.582	0.209	0.005
**Year Level**									
1	Ref.			Ref.			Ref.		
2	0.516	0.317	0.104	−1.529	1.140	0.180	0.103	0.239	0.668
3	0.070	0.375	0.851	−3.498	1.409	0.013	−0.118	0.296	0.690
≥4	−1.006	0.366	0.006	−2.805	1.436	0.051	−0.116	0.289	0.688
**Major**									
Humanities and Social Sciences	Ref.			Ref.			Ref.		
Engineering/Natural Sciences	0.033	0.270	0.904	−0.557	0.938	0.552	0.212	0.207	0.307
Liberal Arts	−0.271	0.414	0.512	−2.041	1.547	0.187	0.184	0.311	0.554
**Grade Point Average**									
≥4.0	Ref.			Ref.			Ref.		
3.5–4.0	−0.088	0.380	0.816	1.267	1.735	0.465	−0.074	0.300	0.804
3.0–3.5	0.675	0.382	0.077	1.827	1.739	0.293	0.159	0.300	0.595
≤3.0	1.125	0.470	0.017	4.352	1.978	0.028	1.099	0.356	0.002
**Allowance**									
Q1 (Low)	Ref.			Ref.			Ref.		
Q2	0.934	0.320	0.004	0.296	1.129	0.793	0.694	0.250	0.005
Q3	1.641	0.361	<0.0001	2.679	1.376	0.052	1.872	0.270	<0.0001
Q4 (High)	1.985	0.373	<0.0001	4.919	1.311	0.000	2.439	0.281	<0.0001
**Living Status**									
Family Home	Ref.			Ref.			Ref.		
Living Alone/Flatting	1.362	0.316	<0.0001	1.775	1.092	0.104	1.279	0.235	<0.0001
College Dorm	0.410	0.318	0.197	−0.469	1.269	0.712	0.868	0.252	0.001
**Smoking Status**									
Current Smoker	1.834	0.343	<0.0001	0.364	1.195	0.761	1.819	0.247	<0.0001
Past Smoker	3.009	0.567	<0.0001	0.812	1.874	0.665	1.850	0.493	0.000
Non-Smoker	Ref.			Ref.			Ref.		
**Underage Drinking**									
Yes	0.671	0.260	0.010	4.163	0.973	<0.0001	2.527	0.199	<0.0001
No	Ref.			Ref.			Ref.		
**Number of sororities/clubs**									
None	Ref.			Ref.			Ref.		
One	0.303	0.270	0.262	1.345	0.960	0.161	−0.165	0.204	0.421
Two or more	1.571	0.434	0.000	1.122	1.394	0.421	0.584	0.344	0.090

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
