# Peer review of "Primary Reason for Drinking Among Current, Former, and Never Flushing College Students"

_ijerph, 2019, doi:10.3390/ijerph16020211_

Round 1

Reviewer 1 Report

Attached as a PDF document.

Author Response

We thank you for giving us the opportunity to revise our paper. In revising our paper, we have carefully considered your comments and suggestions, and done our best to incorporate them accordingly. We appreciate the constructive feedback on our original submission. After addressing the issues raised, we feel that the quality of the paper has improved and we hope you agree. Our responses to each comment are listed below. We have attached revision notes and highlighted the revised sections of the manuscript. Once again, thank you for your valuable and helpful comments.

Attached is a PDF copy of our responses to your insightful comments.

Yours sincerely,

Sarah Oh , Yeong Jun Ju , San Lee , Eun-Cheol Park 

Reviewer 2 Report

Thank you for the opportunity to review this paper.

This is a straightforward research article which is well presented, very clearly written, and easy to follow. I found the rationale for exploring the link between drinking motives and flushing to be strong.

Overall I feel the paper lacks a discussion of the implications of the findings, and this is the main shortcoming of the paper in its current form. I would like to see the authors revisit and expand upon this section greatly, particularly with regard to the “former flushers” group who appear to be placing themselves at significant risk.

Some additional points:

1.     Reasons for drinking: was this recorded as a categorical response, or were categories coded afterwards? If so, what were the processes for coding validation and reliability of coding?

2.     For multiple regression analyses, please report odds ratios and 95% confidence intervals, and overall model fit and R values. Also, please describe how models were structured – were all predictors entered simultaneously?

3.     The significance tests in Figure 1 (discussed lines 166-174); it is unclear what type of test is being reported here.

4.     Table 1: please report column percentages for each category alongside the counts, to allow for easier comparison between flushing groups.

5.     Line 222: I don’t feel that “teetotalism” is the most appropriate term. Suggest use of abstaining from drinking.

Author Response

(The authors gave the same response as above.)

Reviewer 3 Report

I congratulate the authors on an ambitious observational research. The investigation is robust and the design well considered. I look forward to seeing the end result of this study when it is finally complete and published. I commend the authors for their work - both all of the work leading up to this point and for the planning of this study - their contribution to the alcohol related with the health literature. I do have some comments about certain methodological issues covered below under MAJOR ISSUES:

TITLE

The title of this manuscript are a little long. Perhaps a more concise version for clarity, interes and ease of read.

ABSTRACT

It is hard to get the detail in an abstract when the word count is limited and this is often the hardest part of a paper to write. However, I do feel that it would be beneficial to explain what specifically you are looking at in relation to alcohol related with the health (this also applies to the main body of the paper). Is it the development of reasons structural for drinking associated in college students .  This needs to be made clearer throughout the paper.

KEYWORDS:

Please use recognised MeSH terms as this will assist others when they are searching for information on your research topic. The following website will provide these (simply start typing in a keyword and see if it exists or find an alternative if it does not): https://www.ncbi.nlm.nih.gov/mesh

The introduction is weak. An introduction should announce your topic, provide context and a rationale for your work, while catching the reader´s interest and attention. The above has not been given in the introduction that I have read.

Thus, I suggest in this section should be improved, with more details about prevalence, impact related with drinking in college students.

Also, please describe the hypothesis in this section.

MATERIAL AND METHODS:

Please, expand and clarification information related with AUDIT and Flushing tool related with reliability and validity and the actual measurements.

RESULTS:

The results in basis of the used method are correct. Please remove the Figure 1 and describe this information in this section.

DISCUSSION:

Include this section the principal strengths and weaknesses in relation to other studies, discussing important differences in results; the meaning of the study: possible explanations and implications and unanswered questions and future research

CONCLUSION:

These conclusions need to be softened, modified a in order to reflect only the study findings.

Author Response

(The authors gave the same response as above.)

Round 2

Reviewer 1 Report

This study looks at differences in AUDIT scores by participant flushing status and other factors including drinking motivations.  The authors addressed most of my concerns raised in the initial submission.  However, there are still some issues in the revised submission.

Here are my specific critiques and suggestions:

Abstract

Line 14: Change ‘reason’ to ‘reasons’.

Materials and Methods 

Line 115: Change ‘participated in’ to ‘completed’.

Line 115: Not clear if the response rate refers to the 5000 requested participants (where 3435 responded) or 7278 requested participants (where 5000 responded).

Table 1

I am not familiar with the Bonferroni techniques used and the way in which they were reported.  Differences between subgroups are still not able to be ascertained.  I advise cutting p-values and Bonferroni tests from this table (and in the corresponding text).  Once these are cut, Table 1 would still be useful for descriptive results. 

Tables 1 & 2 and Figure 1

When describing the AUDIT outcome variable, label as ‘AUDIT Scores’ (not ‘Alcohol Intake’).  Alcohol intake is only one component of the AUDIT.

Author Response

We thank you for giving us the opportunity to re-revise our paper. In revising our paper, we have carefully considered your comments and suggestions, and done our best to incorporate them accordingly. We appreciate the constructive feedback on our original submission. 

After addressing the issues raised, we feel that the quality of the paper has improved and we hope you agree. Our responses to each comment are listed below. We have attached revision notes and highlighted the revised sections of the manuscript. Once again, thank you for your valuable and helpful comments.

Attached is a PDF copy of our responses to your insightful comments.

Yours sincerely,

Sarah Oh , Yeong Jun Ju , San Lee , Eun-Cheol Park

Reviewer 3 Report

The authors have satisfactorily responded to all of my comments.

Author Response

We thank you for giving us the opportunity to re-revise our paper. Thank you very much for your insightful comments and suggestions. We did our best to incorporate them into our manuscript. 

After addressing the issues raised, we feel that the quality of the paper has improved and we hope you agree. Once again, thank you for your valuable and helpful comments.

Yours sincerely,

Sarah Oh , Yeong Jun Ju , San Lee , Eun-Cheol Park